# Characterization of CD34^+^ Cells from Patients with Acute Myeloid Leukemia (AML) and Myelodysplastic Syndromes (MDS) Using a t-Distributed Stochastic Neighbor Embedding (t-SNE) Protocol

**DOI:** 10.3390/cancers16071320

**Published:** 2024-03-28

**Authors:** Cathrin Nollmann, Wiebke Moskorz, Christian Wimmenauer, Paul S. Jäger, Ron P. Cadeddu, Jörg Timm, Thomas Heinzel, Rainer Haas

**Affiliations:** 1Condensed Matter Physics Laboratory, Heinrich-Heine-University, 40204 Düsseldorf, Germany; cathrin.nollmann@hhu.de (C.N.);; 2Institute of Virology, Heinrich-Heine-University, 40204 Düsseldorf, Germanyjoerg.timm@med.uni-duesseldorf.de (J.T.); 3Department of Hematology, Oncology and Clinical Immunology, Medical Faculty, Heinrich-Heine-University, 40225 Düsseldorf, Germany; paulsebastian.jaeger@med.uni-duesseldorf.de (P.S.J.);

**Keywords:** hematopoietic stem and progenitor cell (HSPC), acute myeloid leukemia (AML), myelodysplastic syndromes (MDS), leukemic stem cells (LSC), CD34, CD38, CD45RA, CD123, PD-L1, flow cytometry, t-SNE, high-dimensional space analyses, classification, dimensionality reduction, immunophenotyping

## Abstract

**Simple Summary:**

Hematopoietic stem and progenitor cells (HSPCs) play a pivotal role in maintaining the homeostasis of the blood and immune systems. Acute myeloid leukemia (AML) and myelodysplastic syndromes (MDS) represent heterogeneous hematologic malignancies resulting from genetic mutations within cells of the hematopoietic lineage, leading to the expansion of leukemic blasts including leukemic stem cells (LSCs). Using the t-distributed stochastic neighbor embedding (t-SNE) methodology, we examined the immunological phenotype of HSPCs based on the differential expression of CD34, CD38, CD45RA, CD123 and programmed death ligand 1 (PD-L1) antigens, and contrasted it with the immunophenotype of blasts and LSCs in AML and MDS.

**Abstract:**

Using multi-color flow cytometry analysis, we studied the immunophenotypical differences between leukemic cells from patients with AML/MDS and hematopoietic stem and progenitor cells (HSPCs) from patients in complete remission (CR) following their successful treatment. The panel of markers included CD34, CD38, CD45RA, CD123 as representatives for a hierarchical hematopoietic stem and progenitor cell (HSPC) classification as well as programmed death ligand 1 (PD-L1). Rather than restricting the evaluation on a 2- or 3-dimensional analysis, we applied a t-distributed stochastic neighbor embedding (t-SNE) approach to obtain deeper insight and segregation between leukemic cells and normal HPSCs. For that purpose, we created a t-SNE map, which resulted in the visualization of 27 cell clusters based on their similarity concerning the composition and intensity of antigen expression. Two of these clusters were “leukemia-related” containing a great proportion of CD34^+^/CD38^−^ hematopoietic stem cells (HSCs) or CD34^+^ cells with a strong co-expression of CD45RA/CD123, respectively. CD34^+^ cells within the latter cluster were also highly positive for PD-L1 reflecting their immunosuppressive capacity. Beyond this proof of principle study, the inclusion of additional markers will be helpful to refine the differentiation between normal HSPCs and leukemic cells, particularly in the context of minimal disease detection and antigen-targeted therapeutic interventions. Furthermore, we suggest a protocol for the assignment of new cell ensembles in quantitative terms, via a numerical value, the Pearson coefficient, based on a similarity comparison of the t-SNE pattern with a reference.

## 1. Introduction

Acute myeloid leukemia (AML) and myelodysplastic syndromes (MDS) are heterogeneous disorders originating from hematopoietic stem cells (HSCs) through the progressive and sequential acquisition of genetic and epigenetic alterations. As a result, there is a clonal expansion of myeloid progenitors/precursors in the bone marrow (BM) and peripheral blood (PB), associated with impaired cell differentiation leading to hematopoietic insufficiency [1,2,3]. Drug resistance and dormancy of the leukemic stem cells (LSC) with a reduced susceptibility to cytotoxic drugs are the main reasons for treatment failure [4,5]. To tackle the problem of resistance and dormancy, a subtle characterization of the leukemic blast population including the LSC is a prerequisite for a more efficacious targeting and eradication.

The search for a better characterization of the various subsets contained within the bulk mass of leukemic blasts from patients with acute leukemia has prompted a constant increase in the number of antigens in panels for single-cell cytometry, reaching numbers from 14 to 28 colors [6,7,8]. In order to visualize and interpret that kind of multidimensional marker expression function (MEF), the traditional representation via two-dimensional scatter plots increases correspondingly and reaches its limits.

Mathematically, these scatter plots represent two-dimensional projections of the multidimensional function, by which some information of the original distribution is inevitably lost. Consequently, rare or so far unknown leukemic subpopulations of pathophysiological relevance may be not detected [9,10] or multi-dimensional structural information may get lost. This prompted the search for two-dimensional, graphical representations of the MEF, which preserve the full representations [11]. Among such mapping algorithms, t-distributed stochastic neighbor embedding (t-SNE) [12] is a promising candidate [13,14,15,16,17]. We opted for the t-SNE algorithm as compared to equally valid alternatives like uniform manifold approximation and projection (UMAP) [18] since it has a long and successful track record over the past decade and is one of the most widely used for comparable tasks [19,20]. Using this methodology, we aimed at elucidating the CD34^+^ cell population in more depth in samples from patients with AML and MDS in comparison to samples from patients in complete remission (CR) following antineoplastic therapy.

## 2. Materials and Methods

### 2.1. Patients

BM samples of 21 patients with MDS (6 patients) and AML (15 patients) were obtained at the Department of Hematology, Oncology and Clinical Immunology from the University Hospital Düsseldorf on their regular follow-up visits for routine diagnostics. Our control population consisted of 12 patients following allografting, cytotoxic chemotherapy or both who were in CR, with 2 patients (#8 and #21 marked with asterisks in Table 1) still not having achieved full hematological reconstitution. The characteristics of the entire group of patients are shown in Table 1.

### 2.2. Isolation and Phenotyping of White Blood Cells

White blood cells (WBCs) were isolated via red blood cell lysis. For that, BM was collected in EDTA coated syringes or blood collection tubes and bone fragments were removed by filtering the BM with a 70 µm cell strainer. The BM was then incubated 1:10 for 10 min with isotonic ammonium chloride solution (155 mM NH_4_Cl, 10 mM KHCO_3_ and 0.1 mM EDTA, pH 7.4, purchased from the University Hospital Düsseldorf Pharmacy, Düsseldorf, Germany). WBCs were pelleted for 5 min at 500 g, supernatant was discarded and the remaining WBCs were washed twice with DPBS prior to staining. For each sample, one to four million cells were transferred to a 96 well U-Bottom plate and dead cells were stained with fixable viability dye (#65-08666-14, Thermo Fisher Scientific, 1:1000 in DPBS), washed with DPBS and subsequently stained for cell surface molecules. Dead cells and surface molecules were each stained for 15 min at room temperature in the dark. Antibodies for cell surface molecules (see Table 2) were diluted in Brilliant Stain Buffer (#566349, BD Horizon, BD Bioscience, Franklin Lakes, NJ, USA) to prevent staining artifacts due to polymer dyes. Prior to data acquisition at a BD LSR Fortessa (V/B/YG/R), cells were washed with DPBS, fixed overnight (IC Fixation Buffer, #00-8222, Thermo Fisher Scientific, Waltham, MA, USA ) and washed again. Cells were taken up in DPBS and acquired at up to 3000 events/s. All samples contain more than 10^5^ cells.

### 2.3. Gating Strategy

In the study presented here, we were particularly interested in a detailed, multi-color flow cytometry-based characterization of the CD34^+^ cells focusing on a subtle comparison between the CD34^+^ cell subsets of patients with active disease (AD) and those of patients in CR. For that purpose, we used a panel of the following monoclonal antibodies: CD34, CD38, CD45, CD45RA and CD123, as it provides the basis for defining the various types of HSPCs (Table 3). In addition, the programmed death ligand 1 (PD-L1) was included, since it is also expressed on normal hematopoietic cells, exerting a suppressive effect on the immunological response. We consider this panel suitable and sufficient to demonstrate strengths and pitfalls of a t-SNE-based analysis.

Our gating strategy for the cells of interest, i.e., the CD34^+^ cells, encompassed six steps including: (1) an FSC vs. SSC gate, (2) a CD45 vs. SSC gate, (3) an exclusion step for the elimination of doublets, and (4) a viability check using eF506 dye for the exclusion of dead cells. As a result, the (5) final gate of interest (GOI) contained CD34^+^ cells excluding the population of granulocytes. Afterwards (6), only the CD34^+^ cells were selected, as shown in Figure 1A and B for patient 1. The gating was carried out in FlowJo^®^ (FlowJo, Ashland, OR, USA). The number of CD34^+^ positive cells of an individual patient varied between 129 and 207,994 (Figure 1C). To avoid domination of features in the t-SNE plots by individual patients, a maximum of 1000 cells was randomly selected in patients with high cell numbers.

### 2.4. Visualization by t-SNE

In a second step, we applied t-SNE for the visualization of different cell clusters based on their similarity with regards to the composition and intensity of antigen expression. The t-SNE algorithm is a nonlinear dimensionality reduction technique which visualizes high-dimensional data in a two-dimensional scatter plot in such a way that the clustering in high dimensions is preserved. Cells exhibiting comparable protein-expression patterns are positioned adjacently on the t-SNE map, facilitating the depiction of distinct cellular subgroups. The nature of this algorithm is to preserve the local relationships and not the global structure [21]. This is one of the well-known limitations of t-SNE [22,23,24]. Thus, global structures such as the arrangement of the clusters and their distances in the t-SNE plane provide no basis for interpretation. A principal component analysis (PCA) was therefore carried out for the initialization to improve the global structure of the plot, as established in the literature [21,25].

To enable comparability between the t-SNE plots, the gated CD34^+^ cells from all patients were first merged into a common data set. The patient ID and the group assignment were appended to the expression matrix prior to data merger permitting the subsequent separation according to these characteristics after the t-SNE analysis. The fluorescence data were scaled biexponentially in a preliminary step [26]. The t-SNE analysis was carried out using the Barnes–Hut implementation of t-SNE by the Rtsne package (Version 0.16, Open Source). The code is available in the Appendix A. For the PCA the predefined value of 50 for the number of retained dimensions was used. The perplexity as well as the number of iterations were varied over wide intervals. These variations, shown in Appendix A, not only reassure us that the structures to be interpreted are robust, but also demonstrate that a perplexity of 70 and 3000 iterations is a reasonable choice providing visibility of the relevant morphology within an acceptable computation time [27]. The t-SNE coordinates (t-SNE1, t-SNE2) were also appended to the expression matrix as novel parameters. The entire data set as well as a data set of only patients in CR and a data set of the patients with AD were then exported as FSC files for further analysis in FlowJo ^®^. The t-SNE plot is created as a function of the two parameters (t-SNE1 and t-SNE2). Since distances within a t-SNE plot cannot be interpreted in a straightforward way for reasons mentioned above, axis labels are omitted for all t-SNE plots, in agreement with common practice.

### 2.5. Defining Gates in the t-SNE Plots

The expression matrix with the t-SNE coordinates of the three data sets (All patients, only CR, only AD) were imported into FlowJo^®^. A group was created therein containing all three data sets. Density-based polygon gates were manually drawn on the common t-SNE plot of all patients in CR. Afterward, the 27 gates were applied to the FlowJo^®^ group in order to transfer them to the remaining two data sets.

### 2.6. Determination of the Immunological Phenotypes of HSPCs

The immunological phenotype of HSPCs was determined using the markers CD34, CD38, CD123 and CD45RA. For this purpose, the limits for the classification into positive (+) and negative (−) according to the marker expression were determined in FlowJo^®^ on the common data set of all patients using scatter plots (Appendix A). The fluorescence values for the markers used were exported separately for the patients in CR and patients with AD for all gates and then displayed as boxplots (Appendix A). The commonly used limits for the classification were drawn into the boxplots with the antigen expression levels and the immunological phenotype of the HSPCs was then determined for the respective gate depending on whether the mean value of the respective marker was above or below the limit.

### 2.7. Quantitative Comparison of t-SNE Plots Using the Pearson Correlation Coefficient

For the quantitative comparison of the t-SNE plots, the density matrix for the respective t-SNE plot was first calculated in R, e.g., for a single patient or for the cumulative image of all patients with AD. The density matrices were exported and the Pearson coefficients between the t-SNE plots were determined using Python. The code is available in the Appendix A. The density plots of all patients are shown in Appendix A.

## 3. Results and Discussion

### 3.1. Design of a t-SNE-Based Protocol for Multicolor Flow Cytometry Analysis

For the t-SNE analysis, a common data set of all FSC files from all patients was created, so that the t-SNE plots are comparable between the patients. The t-SNE analysis was carried out based on the expressions of CD34, CD38, CD45RA, CD123 and PD-L1. In general, the CD34 antigen permits the identification of hematopoietic stem and progenitor cells, while CD38 is considered a marker associated with differentiation [28,29]. The combination of these two antigens with CD45RA and CD123 permits a characterization and quantification within a BM of the HSC/HPC within a BM sample [30,31].

After the t-SNE run on the combined data set (Figure 2A), the contributions of the CR and AD patients were visualized separately to recognize their contributions to the combined t-SNE picture (Figure 2B,C).

By t-SNE, the cells are arranged in five islands (I–V) of different sizes. It becomes immediately apparent that the CR (in 2B) and the AD (in 2C) samples contribute almost complementarily to the combined representation (A). While in the CR patients, most of the cells are in the east part of the main island (I) plus in three of the four separated islands (II, III, V), the cells from the AD patients accumulate more to the west of the main island as well as in island IV. However, the populations are not mutually exclusive, as all cell types are present in both groups, albeit in some regions with strikingly different prevalence. This phenomenon is most likely not related to “contaminating” leukemic cells within the CR samples as the CR patients are MDR negative. In the t-SNE representation, the cells are distributed according to marker-specific gradients, as shown in the bottom row of Figure 2 D–H for the combined data set, where the black horizontal bars in the color scale column define the corresponding intensity intervals. The overlay of the different markers is shown separately for AD and CR in Appendix A, as well as an example for two patients from each of the groups in Appendix A.

As far as CD34 is concerned, the corresponding intensities in (Figure 2D) comprise only positive values since per definition only cells above the threshold of expression were included. Still, the islands in the t-SNE plot show quite varying CD34 expression levels. With respect to CD38, the cells are assorted from northwest to southeast of the main island with increasing expression level, while it is particularly low in island IV. A pronounced CD38 gradient is visible in island III and V, indicating a sub-ensemble of cells undergoing some kind of development. The CD123 concentration, on the other hand, increases from east to west across island I, is almost zero in island II, and shows gradients within islands III and V. The CD45RA expression increases strongly from north to south. Finally, the PD-L1 expression, which is not considered in the assignment of the cells according to Table 3, is non-monotonously distributed across island I and takes characteristic low values in islands II and III.

We can therefore conclude that the gradients in the intensities of CD38, CD45RA and CD123 cause the main substructure in island I, while the expression levels of CD34 and PD-L1 refine this landscape.

### 3.2. Exemplifying Discussion of t-SNE Gates 

For a more detailed study of the five islands, we have defined 27 gates in the t-SNE plot of the CR samples, each with a characteristic set of expression levels for the markers used (Figure 3A). This gate pattern was then transferred without modifications to the AD data as described in Section 2.5, shown in Figure 3B. The percentage distribution of the cells in the 27 gates for the three datasets (all patients, only CR, only AD) is shown in Appendix A.

From that kind of visualization, eight gates emerge, namely gates 1, 6, 7, 10–13 and 15, in which the cells of patients with AD dominate. The remaining gates contain more cells from the CR patients, while within gate 14 the ratio is very close to 1.

The box plots of the 27 gates (Appendix A) were used to assign the cells within each gate according to the classification scheme as detailed in Table 3 and shown in Figure 3F. As can be extracted from Figure 3, the CR subsets are composed of 0.7% HSC/MPP, 9.4% CMP, 1.2% CLP, 11.4% MEP and 44.6% GMP. A proportion of cells (32.7%) could not be allocated according to the classification scheme. On the other hand, the samples of the patients with AD comprised 14.3% HSC/MPP, 4.7% CMP, 4.1% CLP, 13.7% MEP and 47.0% GMP with a proportion of 16.2% of the cells which could not be classified. Clearly, in comparison with the CR samples the AD samples show a significantly greater proportion of HSC/MPP as well as CLP cells, while there are smaller percentages of CMP cells as well as of those cells that cannot be allocated. The fractions of MEP cells are approximately equal for both groups.

Beyond this canonical classification, the t-SNE representation provides a rich substructure within the regions of particular cellular subtypes, reflecting subtle differences between the various populations. A complete delineation of all 27 gates would be certainly beyond the scope of our presentation. We therefore selected gates representing five characteristic cellular subsets, namely gates 1, 3, 6, 12 and 26, to illustrate the possibilities, but also the potential shortcomings associated with a t-SNE representation. The corresponding box plots are shown in Figure 4.

In general, a clear distinction of one gate from the others originates from a particularly low expression of one antigen within this gate. 

We begin with gate 1, a well-separated island containing the great majority of HSC/MPP cells, as defined by the lack, or extremely low expression, of CD38. Since the differentiation between HSC and MPP is based on the CD90 marker (with CD38 negative in both cases) which was not included in our panel, we cannot distinguish these two cell types within our data set. As far as CD45RA and CD123 are concerned, their expression levels show a broad distribution spanning almost the full intensity range. Since AD cells contribute 88% to this population, this gate represents a predominantly leukemic-related gate and is compatible with the signature of leukemic stem cells. We note that the patients’ ID and the group assignment were added to the expression matrix prior to the data merging, which allows us to determine the contribution of each patient group (CR and AD) to each gate. To relate these findings to the results of Kersten et al. [32], we looked at the expression level of CD45RA and CD123 on the CD34^+^ cells within this gate and found a greater expression of these antigens on the leukemic cells compared to those from the control samples. The aforementioned investigators examined the potency of CD45RA to specifically discriminate LSC and normal HSC for a better LSC quantification and found that in comparison to other markers such as CLEC12A, CD33 and CD123, CD45RA was the most reliable antigen. From a clinical point of view, it was interesting to note that CD45RA^+^ LSC tended to be associated with a more favorable cytogenetic/molecular marker constellation. However, it is important to recognize that the expression of CD45RA in AML is not as straightforward as in the immune system T cell subsets, and the functional implications can be quite diverse [33]. With regard to CD123, the study by Testa et al. based on the screening of CD123 expression in various hematopoietic malignancies shows that this antigen not only frequently expressed at high levels in AMLs but also on B-ALLs [34]. In an earlier report, they had explored a large set of AML patients and reported that 45% of these patients overexpress CD123 [35]. Similar to their results, Al-Mawali et al. [36] found that overall, this antigen was expressed in 37 (97%) out of 38 AML cases analyzed. The median expression of CD123 was 90% (range 21%–99%). Interestingly, the proportion of cells co-expressing CD123 on CD34^+^/CD38^−^ leukemic stem cells was also 37 (97%) out of the 38 AML patients with a broad range from 0.0262% to 39.7% (median 0.8164, mean 4.45) at the time of diagnosis. These results are in line with our findings regarding the expression pattern of the CD34^+^/CD38^−^ in our gate 1.

Gate 3, on the other hand, has been selected as an example for a cell cluster mainly encompassing CD34^+^ normal progenitor cells of GMP subtype, as the great majority of cells show a strong CD38 expression in the presence of CD45RA and CD123. Different from this normal signature, the few CD34^+^ cells falling onto this gate from patients with AD are lacking or only faintly expressing CD38 while the intensity for CD45RA and CD123 tends to be stronger in comparison to their normal counterparts. 

Gate 6 resides at the edge of the main island with a proportion of 93% of cells from AD samples. Since the expression levels of all antigens are above the threshold of detection, they are formally classified as GMP. Still, a specific property of gate 6 in comparison to other gates containing GMP-like cells is that the PD-L1 expression level is relatively high—well above the levels in all other gates—and the levels of CD45RA and CD38 are also above the average observed for GMP cells. Furthermore, it is remarkable that these cells have a relatively low CD34 antigen expression and that all antigens display a relatively sharp intensity distribution with relatively low standard deviations. This suggests that there is no ongoing evolution among the cells in this gate. The CD34^+^ cells of this cluster were to some extent CD38^+^, indicating a kind of “late” HSC on its way towards an abnormal stage of differentiation. As far as PD-L1 is concerned, our t-SNE-based data confirm the results obtained previously in a study focusing on the immunophenotype of T cells in patients with MDS and AML [37]. The mechanisms underlying T cell evasion to immune checkpoint inhibitors in acute myeloid leukemia have been recently elucidated by Gurska et al. [38]. 

We now take a closer look at the cells in gate 12, where two-thirds of the cells originate from AD samples. This gate represents a kind of borderline cell pool regarding the AD samples. In general, the expression level of CD45RA is very low, and the CD34 level is extraordinarily high with a relatively broad distribution of CD38 expression. While the cells in this gate from the CR patients are unequivocally classified as CMP, this is not possible for the AD samples, as they rather appear to be a mixture of CMP with HSC/MPP. This gate is, therefore, distinct from most other gates due to its internal shift of the t-SNE intensity between AD and CR samples. Accordingly, the AD cells with a lack or very low expression of CD38 reside more at the left side of this gate, whereas the cells of the CR samples preferentially group around its center. This indicates that the cells undergo an evolution from HSC/MPP when the disease is active, towards CMP during remission. The cluster contained within gate 12 is thus a nice example for the discriminative strength of t-SNE. In comparison to gate 1, the CR group shows a positive CD38 signal, while the AD group in this gate is CD38 negative, even though these values are significantly higher than in gate 1. 

Gate 26 is dominated by a proportion of 68% of CR cells. It is a kind of enigmatic cluster, as this subpopulation of CD34^+^ cells could not be allocated unequivocally according to the classification scheme as described in Figure 3F. Their characterization certainly requires an extended labelling for the lymphoid progenitor cells including antigen markers like CD10, CD7 or CD19. With regard to the leukemic cells contained within this cluster, aberrant marker constellations not related to the canonic scheme are also conceivable. Therefore, starting from our proof-of-principle marker panel, modifications including new monoclonal antibodies are necessary taking into account the steadily evolving knowledge and discovery of leukemic-related antigens and their co-expression patterns. Within this process, our efforts should be geared towards linking the phenotypical characterization to the molecular signature of the leukemic cells in the sense of a phenotype–genotype linkage. In the context of an antigen-targeted therapy, this could be helpful in defining the most relevant subset, i.e., leukemic stem cell, within the bulk mass of leukemic cells. 

We proceed by drawing some general conclusions from these characteristic examples.

First, carefully selected additional markers can discriminate the cells to a deeper level. In that respect, we found a strong correlation between the expression level of the PD-L1 antigen and the percentage of predominantly leukemic cells in a particular gate. Considering that PD-L1 is an immunoprotective antigen, one may speculate that by increasing the PD-L1 expression during the evolution from healthy towards malignant, the cells protect themselves with respect to the immune system.

On the other hand, disregarding a relevant marker can leave the cell population within some of the gates unspecified, as has been seen from the example of gate 26. Moreover, since the markers used tend to show a continuous expression on this cell ensemble, only a few distinct islands became apparent in the t-SNE plot. This means that with manual density-based gating, the areas sometimes do not have a distinct border, which is reflected by the variability of the box plots for the respective gates. By using more markers that ideally exclude each other, better separation within the t-SNE plot [39] may improve subsequent gating or also enable the use of more automated density-based gating, such as DBSCAN [40] or HDBSCAN [41].

Furthermore, in our control samples of patients (CR), the composition of the cell ensemble was similar to our previous findings in normal donors showing a predominance of the GMP followed by the CMP, HSC and the MEP [42]. Subtle differences may be explained by the fact that in our study, BM samples of patients in CR served as normal controls, as BM from normal volunteers were not available. More specifically, normal hematopoietic cells that express high levels of CD34 lacking CD38 are considered stem cells, whereas those that express low levels of CD34 and high levels of CD38 represent more differentiated progenitor cells [43]. The lack of CD38 on leukemic blast cells is also characteristic for the leukemic stem cell [4].

### 3.3. Quantification of the t-SNE Representation 

We proceed by asking to what extent a t-SNE-based assessment can be quantified. Based on quantitative evaluations already proposed [44,45], we suggest an analysis in terms of the Pearson correlation coefficient r (A, B), a well-established measure for the similarity of two pictures labelled A and B. The two pictures are composed of N pixels each, with pixel density A_j_ and B_j_, respectively. The Pearson coefficient is defined as
(1)$$r(A,B)=\frac{cov\left(A,B\right)}{\sigma_{A}\sigma_{B}}$$
with the covariance of the two pictures given by
(2)$$cov\left(A,B\right)=\frac{\sum_{j=1}^{N}A_{j}B_{j}-\left(\sum_{j=1}^{N}A_{j}\right)\cdot \left(\sum_{j=1}^{N}B_{j}\right)}{N}$$
and the standard deviation of the pixel densities of picture X (X = A, B) given by
(3)$$\sigma_{X}=\sqrt{\frac{\sum_{j=1}^{N}X_{j}^{2}-{\left(\sum_{j=1}^{N}X_{j}\right)}^{2}}{N}}$$

For r (A, B) = 1, the two pictures are identical, and they are maximally different, i.e., their sum picture has a density of zero at all sites, for r (A, B) = −1.

The comparison of the two representations of the combined data sets, Figure 2B,C gives $r_{\sum CR,\sum AD}$ = 0.46. Here, ∑AD and ∑CR denote the sum pictures of all AD and all CR samples, respectively. 

Based on this value, we evaluate a classification protocol in which the t-SNE representation of a new sample is generated by first merging it with a reference plot composed of a sufficient number of samples, which is split up again into the two reference pictures ∑AD and ∑CR plus the contribution from the new sample labelled as N.

When we refer to the sample N including its classification, we label it by NCR or NAD, respectively. In the next step, the Pearson coefficients of N with the two reference pictures $r_{\sum AD,N}$ and $r_{\sum CR,N}$ are computed.

We have implemented this protocol with the present data set as follows. From our data set, we have removed each sample separately and considered the remaining combined pictures as reference pictures. We now treat the individual sample N as unknown and compute $r_{\sum AD,N}$ as well as $r_{\sum CR,N}$. This implementation is represented schematically in Figure 5.

The results are the values without parentheses listed in Table 4. For all but one of the twelve control samples, we measure $r_{\sum CR,NCR}$ > $r_{\sum AD,NCR}$, with differences up to 0.47 for sample 20. Therefore, only in the case of *N* = 18, the sample would have been classified as AD in contrast to the correct classification. We will elucidate the reasons for the wrong classification below.

Regarding the identification of an AD sample, the situation is less clear. While samples 1, 12, 13, 14 and 16 show $r_{\sum AD,NAD}$ > $r_{\sum CR,NAD}$ and are thus classified correctly, we observe that $r_{\sum AD,NAD}$ is just slightly smaller than $r_{\sum CR,NAD}\;$in samples 10 and 17, but find dramatic deviations from the classification for samples 4 and 11, with r value differences of 0.68 and 0.42, respectively.

In order to investigate how stably the classification works with respect to multiple t-SNE runs, two further runs were performed, and the classification was carried out as described previously. The AD samples were assigned to the same group in all runs as described above. In two out of three runs, all CR samples except *N* = 18 were identified as CR samples and in the third run, all were classified as CR.

The failure of allocating samples 4 and 11 asks for refined consideration. Despite their different subtype of AML, the leukemic cells show a monoblastic differentiation reflecting a more “mature” subtype not necessarily reflected by a particular CD34/CD38 subset. Since the cells of these misassigned patients represent a more mature type and the patients had only a molecular relapse, it is very likely that they could not be adequately assigned, since the leukemic cells were not contained within the CD34^+^ cell population. For the detection of that kind of subtype, additional markers such as CD33 and CD14, for example, would still be necessary. Rather, the antigen markers used should show expression levels quite similar to those of the CR samples. Since it is of great interest to study how such a misallocation influences the t-SNE representation and the corresponding Pearson coefficients, we remove patients 4 and 11 from the ensemble and repeat the quantitative analysis. The modified t-SNE plot in comparison to the plots of these two patients, shown in Figure 6, illustrates the dissimilarity of the density distributions. The obtained r values are given in Table 4 in parentheses. First, we notice a striking decrease of $r_{\sum AD,\sum CR}$ by 0.22. Apparently, these two samples have been responsible for a significant similarity between the two t-SNE representations, again indicating that samples 4 and 11 generate a pattern that resembles more CR samples than AD cases. Second, for all control samples, the values of $r_{\sum AD,NCR}$ improve, some of them dramatically, e.g. for patient 9, r drops from 0.24 to −0.05. Third, however, we observe some effect on the $r_{\sum AD,NAD}$ values, which change by no more than 0.19. It increases only for patients 1 and 13 but decreases for the remaining cases. This impressively shows how the lack of a relevant marker for clear characterization can lead to false similarities and thus impede the classification. It is therefore conceivable that with each additional diagnostically relevant marker the characterization becomes better, the t-SNE image becomes more differentiated and thus the classification becomes more reliable.

As an evaluation of this proposed identification protocol, we note that the values of $r_{\sum CR,NCR}$ are large, a fact which quantifies the high similarity of the cell population in the CR stage. They are furthermore significantly larger than $r_{\sum AD,NCR}\;$and we can thus conclude that the state of CR is safely identified and clearly distinguished from the AD state. Furthermore, it can be characterized by a single number with the t-SNE-based protocol, namely by $r_{\sum CR,N}$. The identification of an AD case, however, has remained ambiguous. All values for $r_{\sum AD,NAD}$ and $r_{\sum CR,NAD}\;$are close to zero, with correspondingly small differences which in some cases would even indicate a remission. This situation reflects, in our opinion, the heterogeneity of the considered AD cases. Since these samples generate widely varying t-SNE patterns, they have relatively low r values and if such a reference pattern is compared with a new sample, a t-SNE-based identification is ambiguous if not impossible. On the other hand, we have seen in the example of patients 4 and 11 how the t-SNE-based identification can be improved considering blasts of a more “mature” type. We therefore expect that sufficiently differentiated t-SNE reference maps for AD subtypes will also allow the unique identification of an AD as well as its predominant blast population. To be on the safe side, we estimate that t-SNE subtype reference maps should be constructed from at least ten samples.

## 4. Conclusions

Our study has evaluated the potential of t-SNE to represent multi-dimensional cell ensemble data from AML and MDS patients in a compact, two-dimensional form, thereby condensing the widely spread information of the scatter plots in a single picture. In order to develop this mapping into a diagnostically valuable tool, the mapping has to be capable of handling the specific challenges of such data sets, namely the large variance of the cell numbers per sample, the diverse manifestations of the diseases and their subtypes, as well as the unavoidable smearing of the map by the continuous cell evolution. Our protocol takes these initial conditions into account providing a meaningful clustering with gates containing diagnostically relevant cell populations. Additional markers may facilitate further dissection of otherwise homogeneous cell populations or hidden subtypes including rare cells not detectable in the two-dimensional scatter plots. We have also demonstrated how new samples may be diagnosed with the help of reference t-SNE patterns, based on similarities or dissimilarities, respectively. A quantified approach may comprise a statistical measure of the similarity of two pictures, like the Pearson coefficient. Our study shows that such an approach may work even for relatively poorly defined reference pictures. It is straightforward to adapt this concept to the evolution of the cell population of individual patients under therapy.

To facilitate the continued advancement of the t-SNE method, it is essential to establish a consistent and unchanging assignment of cells to a predefined t-SNE map representing “normality,” as detailed earlier. This reference map should remain constant regardless of the introduction of new samples. Recent developments by Kobak, D. and Berens, P. [21] and Policar et al. [46] have introduced methods for embedding new samples into an existing t-SNE plot in single-cell transcriptomics data. The integration of corresponding tools in the mapping algorithm was, however, beyond the scope of the present work and will be the topic of future studies. Finally, it should be emphasized that these concepts are not limited to AML/MDS cells but can be applied to essentially all multidimensional diagnostic fluorescence flow cytometry data.

## Figures and Tables

**Figure 1 cancers-16-01320-f001:**
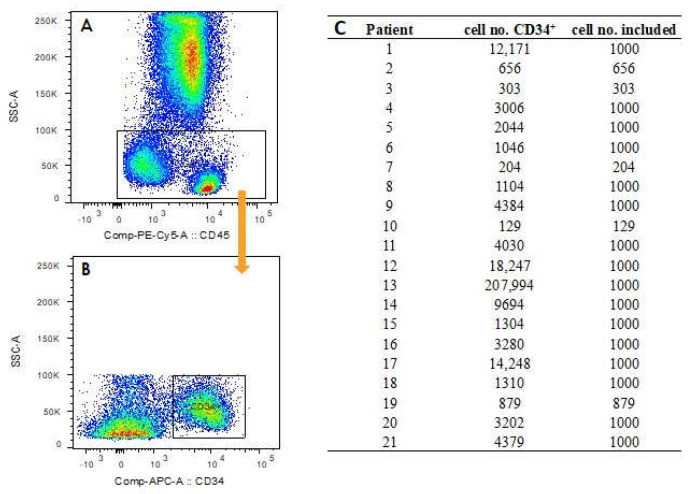
In (**A**) and (**B**), steps (5) and (6) of the gating strategy are shown. The orange arrow means that in B only the cells from the gate of interest (GOI, black frame) were analyzed. (**A**) The exclusion of the population of granulocytes with (5) a GOI; (**B**) The final step (6), selection of the CD34^+^ cells; In (**C**), the numbers of CD34^+^ cells after the gating are shown for each patient as well as the numbers thereof included in the t-SNE analysis.

**Figure 2 cancers-16-01320-f002:**
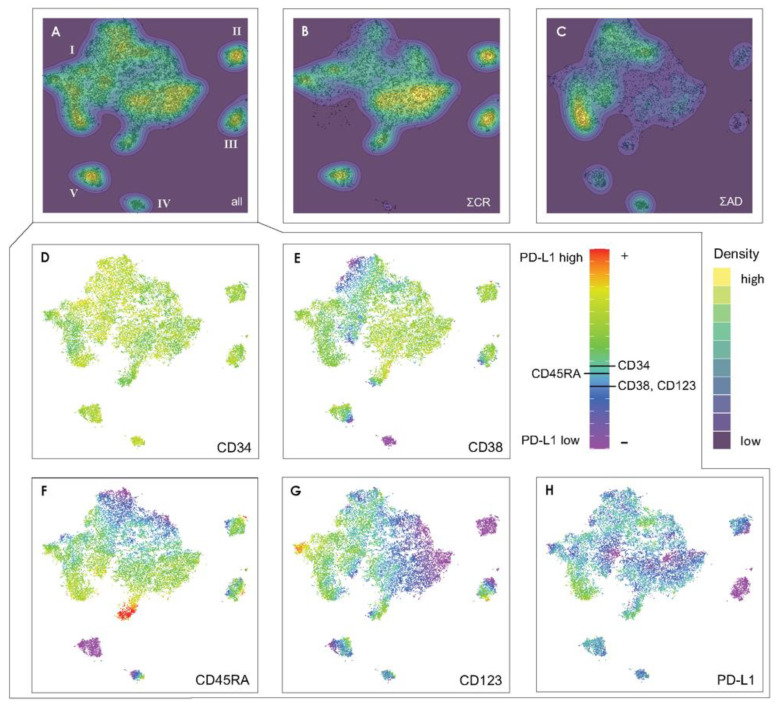
(**A**) t-SNE representation of the combined data set (cells of patients with active disease (AD) and in complete remission (CR)), and the contributions from patients in CR (**B**) as well as from the patients with AD (**C**). The color scale for (**A**–**C**) corresponds to the local density of cells in arbitrary units. In each t-SNE plot, the color scale starts at zero and is normalized to the maximum density in the respective plot. In (**D**–**H**), the prevalence of the various markers entering the t-SNE algorithm are reproduced, for CD34, CD38, CD45RA, CD123 and PD-L1. The color scale represents the expression level. The classification in terms of *positive* (+) and *negative* (−) expression is indicated by the black horizontal lines in the color bar.

**Figure 3 cancers-16-01320-f003:**
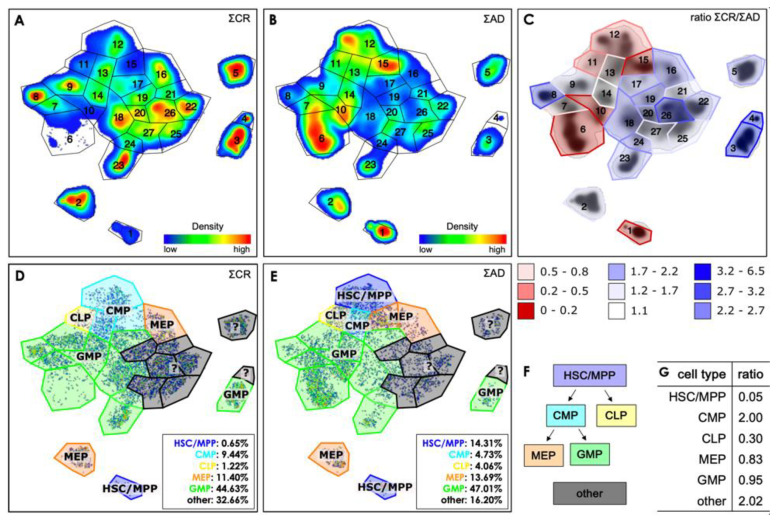
(**A**) Density-based gate definition on the t-SNE plot of the CR data set, performed by visual inspection; (**B**) Application of the gates on the AD data set; In (**C**), the ratio of the percentage distribution of the cells for the CR to the AD samples is given for each gate; The gates are inked in red if this ratio is smaller than 1, i.e., most of the cells in this gate come from AD samples, and blue for ratios larger than 1. The cell type identification for the gates is represented in (**D**,**E**) for the CR samples and for the AD samples, respectively; (**F**) Evolution scheme for the relevant cell types; The ratios of the percentage distribution of cells for the different types in CR samples vs. AD samples are listed in (**G**).

**Figure 4 cancers-16-01320-f004:**
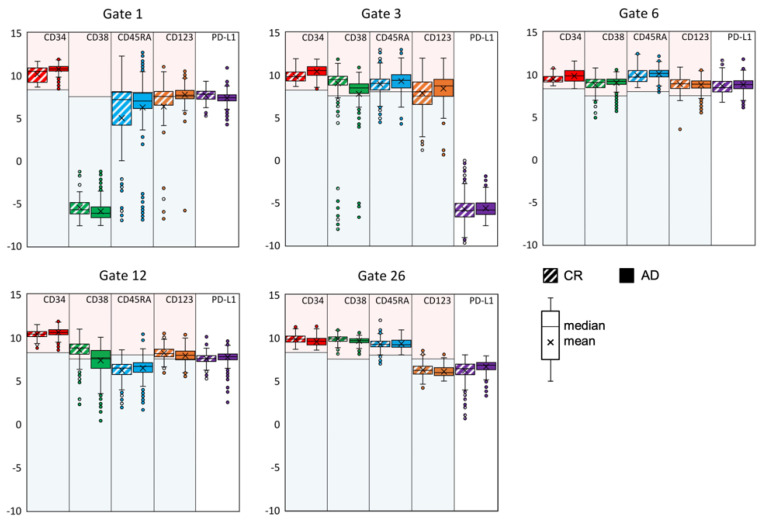
Box plots of the antigen expression levels (red: CD34, green: CD38, blue: CD45RA, orange: CD123, purple: PD-L1) within 5 selected gates from the 27 shown in Figure 3. The fluorescence data were scaled biexponentially in a preliminary step. The light blue and red background indicate the expression levels classified as *negative* (blue) and *positive* (red), respectively. The PD-L1 antigen was not used for the cell classification.

**Figure 5 cancers-16-01320-f005:**
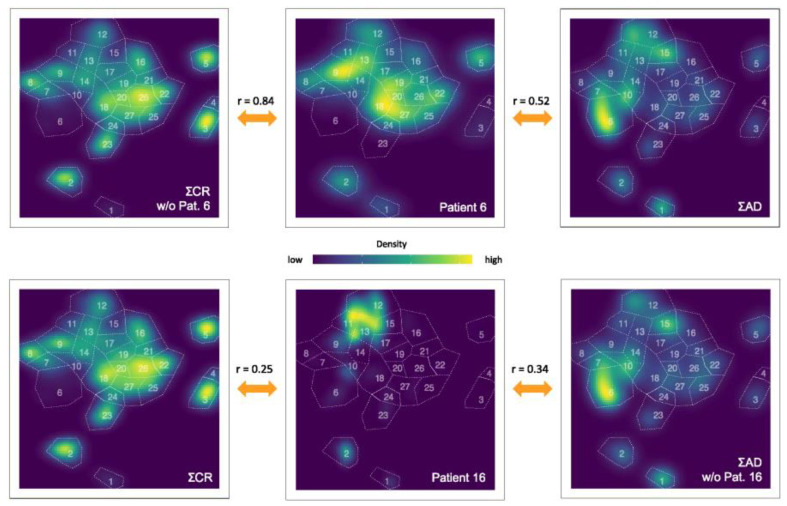
Graphic representation of the calculation of the r value as an example for two patients, 6 (CR group, top center) and 16 (AD group, bottom center). For patient 6, the density distribution is compared with those containing all patients with AD (top right) and for all patients of the CR group except patient 6 (top left). Likewise, we compare for patient 16 the density with the t-SNE plots containing all patients with AD except patient 16 (bottom right) and all patients of the CR group (bottom left). The orange arrows indicate which two plots were compared with each other. The r values are given in between the compared plots. The density is normalized to the respective maximum value of the plot. The gates are shown as an overlay for all plots.

**Figure 6 cancers-16-01320-f006:**
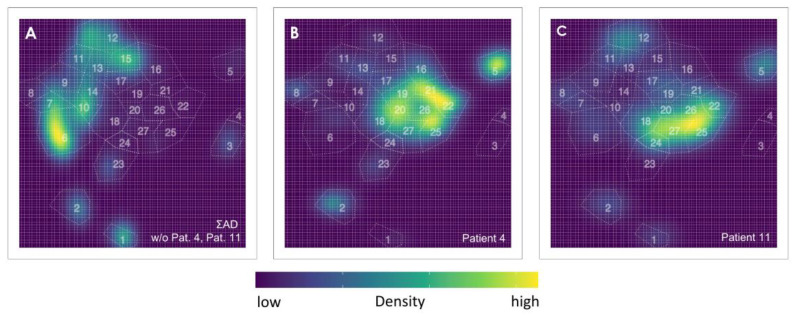
(**A**) Density plot of all patients with AD without patient 4 and patient 11; (**B**) density plot of patient 4, and (**C**) of patient 11. The gates are shown as an overlay in all plots.

**Table 1 cancers-16-01320-t001:** Patient characteristics. The samples are grouped according to whether the patients have active disease (AD) or are in complete remission (CR). All patients in CR are MRD negative (see Appendix A for the MRD analysis).

Group	Pat. ID	Age	Sex	WHO Classification	Status of Disease	Initial Mutation	Cytogenetic	Time ** (Months)
AD	1	61	m	MDS-IB2	AD	-	46, XY	11
AD	4	58	f	AML	Mr	-	47, XX, +11	10
AD	10	58	m	MDS-IB2	Hr	-	46, XY	30
AD	11	67	m	AML, md-r	Mr/p	*FLT3-ITD*, *RUNX1*, *EZH2*	46, XY	5
AD	12	65	f	AML, md-r	Hr	*ASXL1*	46, XX, del(11)(q21,q24) [21]	26
AD	13	60	m	AML	Id	*IDH1*	47, XY, +8[22]/46, XY[2]	0
AD	14	69	f	AML, md-r	Hr	*JAK2*	45, XX, -7	39
AD	16	60	m	AML, md-r	Hr	*ASXL1*, *RUNX1*	not initial: 46, XY, del(3)(q21q25)[23]/47idem+8[5]	53
AD	17	60	f	AML with minimal differentiation	Hr	*IDH2*	47, XX, +mar[4]/46, XX [22], cytogenetic aberration: 7(4;12)	216
CR	2	76	m	MDS-IB2	CR	*ASXL1*	46, XY	13
CR	3	67	f	AML with CEBPA mutation	CR	*CEBPA*	46, XX	21
CR	5	56	f	AML with maturation	CR	*DNMT3A, IDH1*	46, XX	4
CR	6	41	m	AML, md-r	CR	*RUNX1*	complex karyotype	3
CR	7	54	f	AML with NPM1 mutation	CR	*NPM1*, *IDH2*	46, XX	8
CR	8	51	m	MDS with low blasts and SF3B1 mutation (MDS-SF3B1)	CR *	*JAK2*, *SF3B1*	complex karyotype	60
CR	9	67	f	AML with CBFB-MYH11 fusion	CR	*CBFB-MYH11*	46, XX, inv(16)(p13q22)[24]/ 46, XX [3]	29
CR	15	28	f	AML, md-r	CR	*RUNX1*	complex karyotype	47
CR	18	39	f	AML, md-r	CR	*FLT3-ITD*	del(7)(q22[22]/46, XX [3]	2
CR	19	40	m	AML, md-r	CR	*ASXL1*, *c-KIT*, *TET2*	+8, XXY, add(21p)	32
CR	20	61	f	AML, md-r	CR	*ASXL1*, *RUNX1*	46, XX	57
CR	21	70	m	AML, md-r	CR *	*ASXL1*, *RUNX1*, *TET2*, *EZH2*	46, XY	11

* Patients who still not having achieved full hematological reconstitution; ** time difference between initial diagnosis and sample collection; md-r: myelodysplasia-related; CR: complete remission; Hr: hematological recurrence; Mr: molecular recurrence; Mr/p: molecular recurrence/persistence; Id: initial diagnosis; AD: active disease; MDS: myelodysplastic syndrome.

**Table 2 cancers-16-01320-t002:** WBC staining panel.

Specificity	Clone	Fluorescence Dye	Vendor	Cat #	RRID	Concentration
Fixable viability dye	/	eFlour506	TFS *	65-0866-14	/	1:1000
PD-L1	MIH5	PerCP-eFlour710	TFS	46-5983-42	AB_11041815	1:50
CD123	6H6	PE	TFS	12-1239-42	AB_10609206	1:100
CD45	HI30	PE-Cy5	BioLegend	304010	AB_314398	1:200
CD45RA	HI100	PE-Cy7	TFS	25-0458-42	AB_1548774	1:200
CD34	4H11	APC	TFS	17-0349-41	AB_2016604	1:50
CD38	HIT2	APC-eFlour780	TFS	47-0389-41	AB_11217871	1:50

* TFS: Thermo Fisher Scientific.

**Table 3 cancers-16-01320-t003:** Antigen combinations for HSPC characterization.

Cell Type	Label	Antigen Combination
Hematopoietic stem cells	HSC	CD34^+^ CD38^−^ (CD90^+^ not included)
Multipotent progenitor cells	MPP	CD34^+^ CD38^−^ (CD90^−^ not included)
Common lymphoid progenitors	CLP	CD34^+^ CD38^−^ CD45RA^+^
Common myeloid progenitors	CMP	CD34^+^ CD38^+^ CD45RA^−^ CD123^low^ *
Megakaryocyte/erythroid progenitors	MEP	CD34^+^ CD38^+^ CD45RA^−^ CD123^−^
Granulocyte-macrophage progenitors	GMP	CD34^+^ CD38^+^ CD45RA^+^ CD123^+^
Not identified by this set of antigens	Other	Various combinations

* By the term “low” we refer to weakly positive.

**Table 4 cancers-16-01320-t004:** The calculated r values are shown, in the second respective third column, the density of the t-SNE plot of the individual patient (first column) with AD is compared with the density of the t-SNE plot containing all patients with AD (∑AD) respective of the density of the t-SNE plot containing all patients of the CR group (∑CR). In columns 4 to 6, this is shown accordingly for the individual patients from the CR group (fourth column), in each case compared to the density of the t-SNE plot of the entire CR group (∑CR) or entire group of patients with AD (∑AD). In the seventh column, the r value for the comparison between the densities of the two t-SNE plots with all patients from the CR group (∑CR) and with all patients with AD (∑AD) is shown. The calculated r values from the analysis in which patients 4 and 11 were excluded from the AD group are given in parentheses in the respective column.

1	2	3	4	5	6	7
Pat.	NAD vs. ∑AD	NAD vs. ∑CR	Pat.	NCR vs. ∑CR	NCR vs. ∑AD	∑CR vs. ∑AD
*1*	0.23 (0.25)	0.17	2	0.46	0.17 (0.00)	0.46 (0.24)
*4*	0.12	0.80	3	0.53	0.41 (0.36)	
*10*	−0.01 (−0.2)	0.06	5	0.77	0.44 (0.23)	
*11*	0.29	0.71	6	0.84	0.52 (0.31)	
*12*	0.14 (0.13)	0.12	7	0.50	0.43 (0.41)	
*13*	0.05 (0.07)	−0.06	8	0.56	0.19 (0.02)	
*14*	0.22 (0.19)	0.14	9	0.61	0.24 (-0.05)	
*16*	0.34 (0.33)	0.25	15	0.70	0.55 (0.43)	
*17*	0.12 (0.11)	0.15	18	0.43	0.48 (0.42)	
			19	0.28	0.18 (0.16)	
			20	0.67	0.25 (0.01)	
			21	0.69	0.43 (0.28)	

## Data Availability

Data can be provided by the authors upon reasonable request.

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
