# Peer review of "Characterization of CD34^+^ Cells from Patients with Acute Myeloid Leukemia (AML) and Myelodysplastic Syndromes (MDS) Using a t-Distributed Stochastic Neighbor Embedding (t-SNE) Protocol"

_cancers, 2024, doi:10.3390/cancers16071320_

Round 1

Reviewer 1 Report (Previous Reviewer 2)

Comments and Suggestions for Authors

The manuscript significantly improved in the revision process.

It can be published in its current form.

Author Response

Reviewer 1 has no further comments. We thank him / her for the work.

Reviewer 2 Report (Previous Reviewer 4)

Comments and Suggestions for Authors

Nollmann et al. demonstrate in this paper that a t-SNE based analysis of bone marrow cells with multi-color FACS may be useful for discrimination between leukemia (AML or MDS) and non-leukemia patients.  The results are interesting and conceivable. However, there are a couple issues the authors need to address.

1. As shown in Figure 3, the authors could define 27 distinct clusters.  Then the authors identified some clusters as well-defined populations based on cell surface phenotypes indicated in Table 3.  My simple question is then why two distinct Gate 1 and Gate 12 in Figure 3E were marked as the same population, HSC/MPP population.  If you look at the data in Figure 4, CD38 is negative in Gate 1, and positive, at least higher in Gate12, which mean Gate1 is more like HSC and Gate 12 is more like MPP.  Please make any logical explanation how each population was defined.

2. Related to the comment pointed above, Gate 12, which was identified as HSC/MPP in Figure 3E, is defined as CMP in Figure 3D.  I also need a logical explanation about this.

3. In the text, the authors often mentioned leukemic cells, leukemic blasts, or leukemic stem cells without any explanations of how the authors identified such leukemia related cells.  For example, in page 11, line 24, the authors mentioned "the percentage of leukemic cells in a particular gate."  I wander how the authors knew which gates included leukemic cells.  The authors also need to explain how they determined the number of leukemic cells.

4. I believe not all readers who are interested in this paper are familiar with a t-SNE protocol.  Therefore, the authors should modify the text so that more readers will be able to understand the results and interpretation of the data in this study.

Author Response

Dear Editors of Cancers, dear Sir, Dear Madam,

We are glad that reviewer 1 has no further concerns and recommends publication of our manuscript in its present form. We appreciate the feedback of Reviewer 2 and have addressed his/her points in the revision and in this response letter. We believe that by these revisions, our manuscript should be now more accessible and comprehensible to a broad audience.

In the following, please find our responses to the four points raised by reviewer 2. The corresponding modifications in are provided in red color in the revised manuscript.

  1. As shown in Figure 3, the authors could define 27 distinct clusters.  Then the authors identified some clusters as well-defined populations based on cell surface phenotypes indicated in Table 3.  My simple question is then why two distinct Gate 1 and Gate 12 in Figure 3E were marked as the same population, HSC/MPP population.  If you look at the data in Figure 4, CD38 is negative in Gate 1, and positive, at least higher in Gate12, which mean Gate1 is more like HSC and Gate 12 is more like MPP.  Please make any logical explanation how each population was defined.

The immunological phenotypes of the HSPCs were determined as described in section 2.6 of the manuscript. Classification into positive and negative categories was conducted based on scatter plots (refer to Supplement Fig. S2), with subsequent application of defined limits to the box plots (see Supplement Fig. S3), performed separately for the CR and AD groups. The respective phenotypes were assigned based on whether the mean value exceeded or fell below the specified limit. In particular, for gate 12, the CR group exhibited CD38 positivity, while the AD group showed negativity. Even though values were elevated compared to Gate 1, the mean value for the AD group remained below the limit. It is noteworthy that the differentiation between HSC and MPP relies on the CD90 marker, with both cell types being CD38 negative. As CD90 was not included in the panel, we cannot make a definitive statement regarding this distinction could not be made.

We have elaborated on this issue in the revision by adding the following paragraphs in 3.2 at the discussion of gate 1 and at the end of the discussion of gate 12:

“Since the differentiation between HSC and MPP is based on the CD90 marker (with CD38 negative in both cases) which was not included in our panel, we cannot distinguish these two cell types within our data set.”

“In comparison to Gate 1, the CR group shows a positive CD38 signal, while the AD group in this gate is CD38 negative, even though these values are significantly higher than in Gate 1.”

  1. Related to the comment pointed above, Gate 12, which was identified as HSC/MPP in Figure 3E, is defined as CMP in Figure 3D.  I also need a logical explanation about this.

Figure 3D displays the assigned cell types for the CR group, while Figure 3E shows the assigned cell types for the AD group. As explained in response to point 1, the cells in gate 12 were CD38 positive for the CR group and negative for the AD group. Therefore, Gate 12 is assigned as CMP (CD34+CD38+CD45RA-CD123low) for the CR group and as HSC/MPP (CD34+CD38-) for the AD group. This should now become clearer with our explanation in the revision in response to point 1.

  1. In the text, the authors often mentioned leukemic cells, leukemic blasts, or leukemic stem cells without any explanations of how the authors identified such leukemia related cells.  For example, in page 11, line 24, the authors mentioned "the percentage of leukemic cells in a particular gate."  I wander how the authors knew which gates included leukemic cells.  The authors also need to explain how they determined the number of leukemic cells.

This study presents an analysis of bone marrow samples obtained from patients with a myeloid stem cell disorder. Of the patients, nine (AD group) have an active disease, with seven of them exhibiting more than 5 percent blasts in the bone marrow based on morphological criteria. For the other two patients with molecular relapse, it cannot be ruled out that the CD34+ population also contains some normal hematopoietic progenitor or stem cells. Considering this aspect, we use the term “predominantly leukemic cells” in the text. For semantic clarity the expression leukemic cells is used throughout the entire text replacing “leukemic blast, leukemic stem cells”. As the patient ID and group assignment were added to the expression matrix before data merging, it is possible to determine the percentage composition of each gate with respect to the CR and AD groups (Supplement Table S1), as explained in section 2.4. By knowing the patient group and specific phenotypes, it is possible to identify predominantly leukemic cells.

In order to present our identifications more comprehensibly, we have added the following two sentences to the discussion of Gate 1 in 3.2:

“We note that the patients’ ID and the group assignment were added to the expression matrix prior to the data merging, which allows to determine the contribution of each patient group (CR and AD) to each gate.”

  1. I believe not all readers who are interested in this paper are familiar with a t-SNE protocol.  Therefore, the authors should modify the text so that more readers will be able to understand the results and interpretation of the data in this study.

In considering your final suggestion, we acknowledge the importance of ensuring accessibility to a broad readership, including those who may not be familiar with specific details of a t-SNE protocol. As outlined by the editors of “Cancers” in the chapter “AI in Medical Imaging and Imaging Processing” the scope of papers - such as ours - falling into this particular section is to present the current knowledge dedicated to AI methods used in medical systems, with their applications in different fields of diagnostic imaging. With this conceptual background in mind, we were aware of the inherently heterogeneous audience, still assuming that the majority of our readership would be familiar with topics and works at the interface of these two domains. Furthermore, given the richness of our findings, we were compelled from the outset to optimize the use of available space, balancing comprehensive coverage of fundamental methodological aspects with sufficient room for data presentation. Following that line of reasoning, we provided - in addition to a more general introduction to t-SNE - a detailed description of all methodological steps and explanation of the pertinent parameters in section 2.4. For broadening the understanding of the procedure, the code is also available in the supplement. Beyond the “Material and Method” section, the first part of the “Results and discussion” section also describes the avenues towards the formation of the various islands, which are based on the marker intensities. For readers requiring a more foundational understanding of t-SNE methodology, recourse to the literature references provided in the introduction and discussion sections offers additional access to deeper understanding.

Furthermore, it should be recognized that there can never be a one-size-fits-all version of a manuscript, given the varying levels of expertise among readers. Finally, we note that none of the previous five referees has considered our introduction and explanation of t-SNE incomprehensible.

To summarize and in balancing the different aspects as outlined above, we feel that readers interested in the details of t-SNE and our specific implementation thereof should be able to collect the corresponding information based on the review articles we cite, plus the supplementary information provided.

Yours sincerely, on behalf of all authors,

Thomas Heinzel

This manuscript is a resubmission of an earlier submission. The following is a list of the peer review reports and author responses from that submission.

Round 1

Reviewer 1 Report

Comments and Suggestions for Authors

1. Summary

Nollmann et al. investigate the cellular composition of the immature, CD34+ CD38- blast fraction in AML and MDS patients using t-SNE. The manuscript contains some interesting aspects and is timely since computational techniques to analyze thev increasingly high-dimensional flow data sets need to be developed urgently. However, I have some reservations and comments that need to be addressed before a potential publication.

2. Clarity and context

The manuscript is overall clearly written, and the goals are clear. The reader can follow the course of analysis and conclusions drawn from the data. Conclusions are generally supported by the data shown and the figures have sufficient quality.

3. Major points

Materials and methods are not outlined in sufficient detail to reproduce or understand many of the findings.

1) The R script used to generate the results should be made available to the readers/reviewers via GITHUB or as a supplementary file.  

2) For the computational analysis many critical parameters, such as the numbers of cells, the perplexity parameter, or the iterations of the algorithm and the antigens used seem to have been set arbitrarily without any control whether the results of the analysis are stable when slightly other values for these parameters would have been chosen. The number of dimensions that were retained from the initial PCA and served as input into t-SNE is not mentioned at all. At least 3-4 other numerical values for each of these parameters should be tested to demonstrate stability of the t-SNE embedding. The reasons why the numerical values were chosen as they were should be mentioned, especially when they deviate from the default values of the R-package. The Kullback-Leibler divergence can be used as an objective measure to judge the quality of each t-SNE embedding. Thus, a grid search for the smallest KL-divergence should be performed and it should be shown that the t-SNE embeddings are stable and the one used in the paper does not differ substantially from the solutions obtained when other parameters are chosen (within a reasonable range).   

3) Especially the perplexity parameter is key for the results of the t-SNE visualization (https://distill.pub/2016/misread-tsne/).  Since the value used here (P=70) differs from the value of 5-50 recommended by Maaten and Hinton in their original paper, the influence of this parameter on the visualization should be investigated in more detail to show that the t-SNE plots used in the publication faithfully represent the higher-dimensional data set. 

4) It is not quite clear to me why CD45 and SSC/FSC and other fluorochromes used in the flow panel should not contain any useful information to separate LSC and HSC. The advantage of dimensional reduction techniques such as t-SNE is that we do not necessarily need to narrow the number of investigated parameters down but can use the whole dataset. Irrelevant parameters would not contribute to the separation of the clusters in the high dimensional space and therefore not used by t-SNE, unless there is a great technical variation in them that is unrelated to biology. Please clarify. 

5) The complete flow staining panel including the antibody clones, fluorochromes, vendor must be given. Also, the staining protocol and erythrocyte lysis reagents (if used) and staining buffers must be described in sufficient detail to ensure reproducibility of the study. The instrument on which the samples were acquired must be given.

6) The gating strategy lacks sufficient detail– normally, immature blasts are identified by gating on CD45dim SSC-low blasts followed by selection of a primitive marker (CD34 or CD133 or CD117). The authors also mention the use of CD45 vs SSC plot, but this is confusing, since the complete flow panel was not supplied and the authors only refer to five antibodies in the text, none of which is CD45 (CD34, CD38, CD45RA, CD123 and PD1-L). 

7) In order to get a sense for the variability of the t-SNE visualization, the overlay of the different markers in Figure 2D-H should also performed for the AD and HR separately and also for 3-4 representative samples. (supplement)

8) Manual clustering in the low-dimensional representation is arbitrary and relies heavily on the properties of the t-SNE embedding in 2D space, which, as shown above, depends on the arbitrary choice of t-SNE parameters. Stability should be shown by using different t-SNE embeddings. Better choice would be to compare the manual gating with a density-based algorithm such as ClusterX (Rodriquez and Laio, Science 2014) as implemented in the cytofkit package. As all dimensional reduction techniques invariably lead to loss of information the authors could discuss or even try whether the clusters can be better separated in the high dimensional space ? 

8) It should be pointed out, that patients in hematological remission after intensive chemotherapy or allo-PBSC do not represent healthy samples. When available, MRD (measurable residual disease) information should be included in Table 1. The analysis would benefit from samples from healthy donors, where the ground truth (no AML present) is known. This can be exemplified for Gate 1, which is a disjoint set of points not connected to the normal continuous landscape of differentiating hematopoiesis (unlike gate 12). The cells in this cluster likely represent LSC (leukemic stem cells) as normal CD34+ CD38- HSC do not express CD45RA or CD123 (Zeijlemaker et al. Leukemia 2016). Therefore, the few cells in this cluster originating from HR patients likely represent residual leukemic stem cells. If available, this should be correlated to MRD information (qPCR/NGS) or information on relapse in these patients.

9) The assignment of samples to AD or HR based on global similarity of t-SNE representations using the correlation coefficient is interesting. However, from a clinical point of view there is more need to detect MRD than to assign a sample with frank leukemia (as shown in Figure 1) to “AD”. Therefore, the observation that LSC form distinct clusters (not detectable in healthy donors?) is likely far more important and should be validated in future studies.

 4. Minor points

1) Introduction: “The restriction to five fluorescence markers preserves the full phenomenology of higher-dimensional data …” This statement seems to be unconnected to the rest of the introduction and also not supported by data or references. Please rephrase or clarify.

2) Results 3.1: The first paragraph contains many experimental details, which better belong into the M&M section or the figure caption.

3) “However, the populations are not mutually exclusive, as all cell types are present in both groups…” this could be due to positive MRD in some HR patients, since healthy donors were not investigated. Please clarify/rephrase.

4) The discussion is too long, please focus more on the relevant novel parts of the work.

Reviewer 2 Report

Comments and Suggestions for Authors

Revision of manuscript cancers-2778086, Nollmann et. al., Characterization of CD34+ Cells from Patients with Acute Myeloid Leukemia (AML) and Myelodysplastic Syndromes (MDS) using a t-distributed stochastic neighbor embedding (t-SNE) protocol.

The manuscript attempts to create a new protocol to characterize primary AML and MDS samples by using a limited number of marker proteins. The manuscript clearly elucidates that due to the heterogeneity of the disease development it is a challenge to unambiguously categorize these samples. A combinatory characterization of CD marker proteins with phenotypic cell characterization will remain necessary to definitely classify leukemic situations. Still, the manuscript provides an interesting approach for the classification of CD34+ AML/MDS samples. The discussion and conclusions clearly describe the limits of this protocol. The sentence “a protocol which takes these initial conditions into account and have shown that meaningful clustering is observed, with gates in the t-SNE plot which can, in many cases, be related to diagnostically relevant cell populations” fairly elucidates the limitation of this approach.

The manuscript has some main drawbacks, which should be addressed in the revision process:

-          The system is based on AML and MDS patient samples. I wonder, if the system would be able to outselect non-AML and non-MDS leukemic situations. This should be commented.

-          The basis of the two dimensional data set presentation of the t-SNE representation has to be presented more clearly. What is the x- and y-axis of this two dimensional distribution?

-          Since the data processing is mainly based in the density-based gating strategy on the t-SNE, it should be introduced more clearly.

-          Since the gating profile is essential for the t-SNE setting, the strategy for the gating process in figure 3A should be introduced more detailed. It might help to demonstrate the abundance of the relevant cell types in the particular gates in a suppl. table.

-          The number of patient samples was very small and therefore it is hardly representative for a generalized judgement. I would recommend to increase the number of both cohorts. In addition, it would be important to use this approach for newly diagnosed patient samples.

-          The basis of the data evaluation was the total number of cells and not their percental distribution usually used for flow cytometric data evaluation. Why?

In addition, some minor improvement should be made:

-          Explain “the nature of the algorithm is to preserve the local relationships and not the global structure”. What is meant by “global structure”?

-          As mentioned above, in all figures x- and y- axis of the two-dimensional blots would have to be labelled.

-          Do the blots in figure 2D – 2H include ALL AD and HR samples? I think it would be interesting to demonstrate these blots separate for AD and HR samples in a suppl. figure.

-          According to the description, the sentence “The CD123 concentration, on the other hand increases from east to west across island I, …” is wrong.

-          The labelling of the data presented in the blots of figures 5 and 6 would ease reading the figure. Why were these examples taken?

-          Explain the parentheses in legend table 3. The table should at one page. Label the column numbers.

-          The scaling of the blots in figures S1 is not clear. Enlarge the labels and describe the scaling.

-          Figure S2: Explain the cell type specifications in the bottom of the graph columns.

Reviewer 3 Report

Comments and Suggestions for Authors

In their paper, Nollmann et al investigated the phenotypic data of CD34+ on a cohort of patients with AML, analyzing with advanced modalities the antigen expression along with cell maturation.

The study has merits and is well conducted, the paper well-written. However, there are some issues to be addressed, as detailed below.

Overall, the paper is someway unbalanced, as the authors provide great granularity into the advanced analysis of phenotypic data and their interpretation, whereas the study presents some weaknesses on other sides (biological characterization of AML cases, choice of samples, discussion of findings), that appears the main limitation of the project.

Major issues

- Results: the characteristics of the AML cases is definitely outdated, for instance referring to FAB morphology. The authors have to classify their cases according to updated AML classification as WHO (5th edition) and/or ICC, in case leaving the morphological classification in a separate column of the table, if this information remains important in the mind of the authors. To align with this requirement, the authors have to provide molecular and cytogenetic data, on turn important for a better interpretation of the analysis they carried out by flow cytometry.

- Methods: the authors use bone marrows of patients in remission after chemotherapy as a reference control

- Discussion: Please reformulate the expression “In that respect we found a strong inverse correlation between the expression level of PD-L1 and the “healthiness” of the cells” as “healthiness” does not seem having a scientific expression actually. On the same level, the next sentence “during the evolution from healthy towards malignant, the cells protect themselves”, again conveying naïve information. The authors should mitigate their speculations, merely describing their data, which are potentially interesting, and discussed properly on them. As further instance, I don’t think to define “plausible” the expression of PD-L1 sounds correct in this context; rather, I would describe the finding and speculate on the potential consequences (that is protection from immunological control of leukemic/neoplastic cells).

- Discussion: this section is definitely too long especially as the authors repeat the findings of Results section in a large part of their discussion, that is not useless and deviates from the primary aims of the article.

- Conclusions: definitely too long, this section is supposed to serve conveying the summary of the findings by the authors and should not be the mere repetition of a part of the findings.

Minor issues:

- The definition of “hematological remission” here (“but they may not have returned to completely normal levels”) sounds a little bit naïve or an interpretation of the authors; I would suggest referring to “complete remission” quoting the recognized guidelines for its definition

Comments on the Quality of English Language

Minor editing of English language required

Reviewer 4 Report

Comments and Suggestions for Authors

Nollmann et al. demonstrate in this paper that CD34+ bone marrow cells from patients with AML/MDS can be subdivided more than we have believed based on the results by t-SNE analysis. The results are interesting and informative. But since biological characterization of each population (gate) was not performed in this paper, most of the authors' notions or conclusions might be just speculation. For example, the authors mention that "Gate 6 represents another leukemic related cluster.."  But if Gate 6 is a unique population, the authors should at least examine morphology of the cells in Gate 6 to make sure that Gate 6 is truly a leukemic related cluster. Without such biological analyses, this paper should be more suitable for a special journal in the field of bioinformatics.  

Reviewer 5 Report

Comments and Suggestions for Authors

Review Report Attached
